# Continuous Bose–Einstein condensation

Chun-Chia Chen[1], Rodrigo González Escudero[1], Jiří Minář[2,3], Benjamin Pasquiou[1,3], Shayne Bennetts[1,3] & Florian Schreck[1,3✉]

Bose–Einstein condensates (BECs) are macroscopic coherent matter waves that have revolutionized quantum science and atomic physics. They are important to quantum simulation[1] and sensing[2,3], for example, underlying atom interferometers in space[4] and ambitious tests of Einstein's equivalence principle[5,6]. A long-standing constraint for quantum gas devices has been the need to execute cooling stages time-sequentially, restricting these devices to pulsed operation. Here we demonstrate continuous Bose–Einstein condensation by creating a continuous-wave (CW) condensate of strontium atoms that lasts indefinitely. The coherent matter wave is sustained by amplification through Bose-stimulated gain of atoms from a thermal bath. By steadily replenishing this bath while achieving 1,000 times higher phase-space densities than previous works[7,8], we maintain the conditions for condensation. Our experiment is the matter wave analogue of a CW optical laser with fully reflective cavity mirrors. This proof-of-principle demonstration provides a new, hitherto missing piece of atom optics, enabling the construction of continuous coherent-matter-wave devices.

Continuous operation is advantageous for sensors as it eliminates dead time and can offer higher bandwidths than pulsed operation[9–12]. Meanwhile, sensors using BECs benefit from their high phase-space density and unique coherence properties[2–6,13]. Combining these advantages, a CW atom laser beam outcoupled from a CW condensate could be ideal for many quantum sensing applications[14–17]. In the long term, CW atom lasers could benefit applications ranging from dark-matter and dark-energy searches[18,19], gravitational-wave detection[20–24], tests of Einstein's equivalence principle[5,6] to explorations in geodesy[25–27]. In the short term, the CW BEC offers a platform to study quantum atom optics and new quantum phenomena arising in driven-dissipative quantum gases[28].

The key to realizing a CW BEC of atoms is to continuously amplify the atomic matter wave while preserving its phase coherence[29]. An amplification process is essential to compensate naturally occurring atom losses, for example, from molecule formation. It is also needed to replace the atoms that will be coupled out of the BEC for sustaining an atom laser or atom interferometer. Addressing this challenge requires two ingredients: a gain mechanism that amplifies the BEC and a continuous supply of ultracold atoms near quantum degeneracy.

The first steps towards a continuous gain mechanism were taken in ref. [30], in which merging of independent condensates periodically added atoms to an already existing BEC, but in which coherence was not retained across merger events. A Bose-stimulated gain mechanism into a single dominant mode (the BEC) is required to provide gain without sacrificing phase coherence. Such gain mechanisms have been demonstrated using elastic collisions between thermal atoms[29,31], stimulated photon emission[32], four-wave mixing[33,34] or superradiance[35]. However, in all these demonstrations, the gain mechanism could not be sustained indefinitely.

To sustain gain, the second ingredient is needed: a continuous supply of ultracold, dense gas with a phase-space density—the occupancy of

the lowest motional quantum state—approaching $\rho = 1$. Great efforts were spent developing continuously cooled beams of atoms[36–39] and continuously loaded traps[7,8], which—so far—have reached phase-space densities of $\rho = 10^{-3}$. To achieve the required microkelvin temperatures, these experiments have to use laser cooling, but near-resonant laser-cooling light is highly destructive for BECs[40]. Several experiments have maintained a BEC in the presence of harmful light, either by spatially separating the laser cooling from the quantum gas[7,30,36,39,41] or by reducing the absorbance of the quantum gas[29,42–44].

Here we demonstrate the creation of a CW BEC that can last indefinitely. Our experiment comprises both ingredients, gain and continuous supply, as illustrated in Fig. 1. The centrepiece of the experiment consists of a large 'reservoir' that is continuously loaded with Sr atoms and that contains a small and deep 'dimple' trap in which the BEC is created. The gas in the reservoir is continuously laser-cooled and exchanges atoms and heat with the dimple gas. A 'transparency' beam renders atoms in the dimple transparent to harmful laser-cooling photons. The dimple increases the gas density while the temperature is kept low by thermal contact with the reservoir. This enhances the phase-space density, leading to the formation of a BEC. Bose-stimulated elastic collisions continuously scatter atoms into the BEC mode, providing the gain necessary to sustain it indefinitely.

## Experiment

To continuously refill the reservoir, a stream of atoms from an 850-K oven flows through a series of spatially separated laser-cooling stages. The initial stages load a steady-state magneto-optical trap (MOT) operated on the 7.5-kHz $^1S_0 – ^3P_1$ transition[7], shown in Fig. 1a. An atomic beam of µK atoms is then outcoupled and guided[39] 37 mm to the reservoir. This long-distance transfer prevents heating of the atoms in the reservoir by laser-cooling light used in earlier cooling stages.

[1]Van der Waals-Zeeman Institute, Institute of Physics, University of Amsterdam, Amsterdam, the Netherlands. [2]Institute for Theoretical Physics, Institute of Physics, University of Amsterdam, Amsterdam, the Netherlands. [3]QuSoft, Amsterdam, the Netherlands. ✉e-mail: ContinuousBEC@strontiumBEC.com

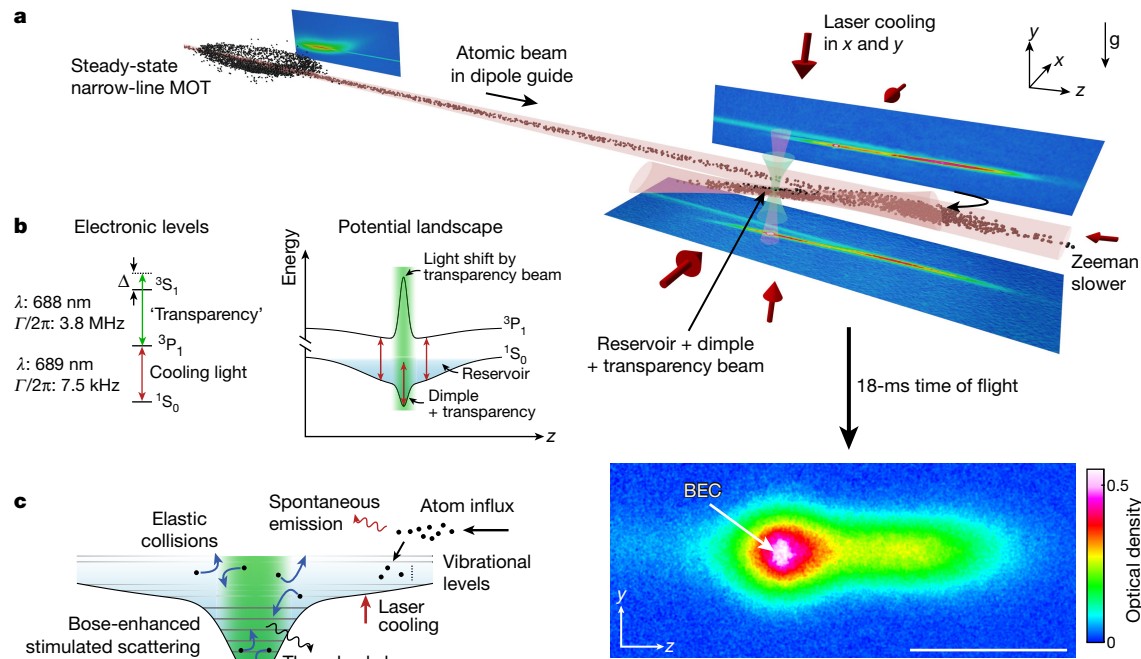

**Fig. 1 | Experimental setup and scheme. a**, $^{84}$Sr atoms from a steady-state narrow-line magneto-optical trap (MOT) are continuously outcoupled into a guide and loaded into a crossed-beam dipole trap that forms a large reservoir with a small, deep dimple. Atoms accumulate in the laser-cooled reservoir and densely populate the dimple, in which a BEC forms in steady state. After time-of-flight expansion, the BEC shows as an elliptical feature in the centre of an absorption image. The scale bar denotes 1 mm. **b**, By off-resonantly

addressing the $^3P_1$–$^3S_1$ transition using a 'transparency' laser beam, we produce a strong, spatially varying light shift on the $^3P_1$ electronic state, rendering atoms locally transparent to laser-cooling photons addressing the $^1S_0$–$^3P_1$ transition. This enables condensation in the protected dimple region. **c**, Schematic of the potential landscape from both reservoir and dimple trap, and of the dominant mechanisms leading to BEC atom gain and loss.

To slow the roughly 10-cm·s$^{-1}$ atomic beam and load it into the reservoir while minimizing resonant light, we implement a Zeeman slower on the $^1S_0 - {}^3P_1|m'_J = -1\rangle$ transition. This slower uses a single, counter-propagating laser beam together with the 0.23-G·cm$^{-1}$ MOT magnetic field gradient along the guide (see Methods). The 11.5-µK-deep reservoir is produced by a horizontal 1,070-nm laser beam focused to an elliptical spot with waists $w_y = 14.5$ µm vertically and $w_x = 110$ µm horizontally. A 6° horizontal angle between the guide and the reservoir allows the decelerated atoms to be nudged into the reservoir after reaching the intersection. The atomic beam and the reservoir are radially cooled by two pairs of beams addressing the magnetically insensitive $^1S_0|m_J = 0\rangle - {}^3P_1|m'_J = 0\rangle$ transition.

This arrangement of traps and cooling beams leads to the loading of the reservoir with a flux $\Phi_R = 1.4(2) \times 10^6$ atoms s$^{-1}$ (see Supplementary Information), a radial temperature of $T_{Rr} = 0.85(7)$ µK and an axial temperature of $T_{Rz} = 3.0(5)$ µK. The corresponding phase-space flux is $\kappa = \left(\frac{\partial\rho_R}{\partial t}\right)_T = \Phi_R\left(\frac{\hbar^3\omega_{Rx}\omega_{Ry}\omega_{Rz}}{k_B^3 T_{Rr}^2 T_{Rz}}\right) = 5.0(2) \times 10^{-2}$ s$^{-1}$ (ref. [45]), in which $\hbar$ is the reduced Planck constant, $k_B$ the Boltzmann constant and $\omega_{Ri}/2\pi$ are the reservoir trap frequencies.

To reduce heating and loss, we use a 'transparency' laser beam[43] that renders atoms in the dimple trap transparent to near-resonant cooling light. This beam is overlapped with the dimple and its frequency is set at 33 GHz blue detuned from the $^3P_1$–$^3S_1$ transition, so as to locally apply a differential light shift on the $^1S_0$–$^3P_1$ transition; see Fig. 1b and Methods. All transitions to the $^3P_1$ manifold are thereby shifted by more than 500 times the $^1S_0$–$^3P_1$ linewidth, whereas atoms in the $^1S_0$ ground state experience a light shift of only 20 kHz. Without the transparency beam, the lifetime of a pure BEC in the dimple is shorter than 40 ms, whereas with the transparency beam, it exceeds 1.5 s (see Methods).

For a BEC to form in the dimple, the ultracold gas must exceed a critical phase-space density of order one. The dimple is produced by

a vertically propagating 1,070-nm beam with a 27-µm waist focused at the centre of the reservoir. In the steady state, the $6.9(4) \times 10^5$ atoms in the dimple are maintained at a low temperature ($T_D = 1.08(3)$ µK) by thermalization through collisions with the $7.3(1.8) \times 10^5$ laser-cooled atoms in the reservoir[43]. The dimple provides a local density boost thanks to its increased depth (7 µK) and small volume compared with the reservoir[46–48]. This leads to a sufficient phase-space density for condensation.

In a typical instance of our experiment, we suddenly switch all laser beams on and let atoms accumulate in the reservoir and dimple for a time $t_{hold}$. The phase-space density in the dimple increases and—eventually—a BEC forms. The BEC grows thanks to preferential Bose-stimulated scattering of non-condensed atoms into the macroscopically populated BEC mode. This produces continuous-matter-wave amplification, the gain mechanism for the CW BEC[31]. The BEC grows until losses eventually balance gain and steady state is reached.

## Analysis of the CW BEC

We now demonstrate the existence of a BEC and later show that it persists indefinitely. To tackle the first point, we analyse atomic cloud density images for $t_{hold} = 2.2$ s and 3.2 s, immediately before and after the formation of a BEC, as shown in Fig. 2a, b. These $x$-integrated absorption images are taken after switching off all laser beams and letting the cloud expand for 18 ms. Both images show broad distributions of thermal atoms that are horizontally extended, reflecting the spatial distribution of the gas before expansion. Notably, the image for the longer $t_{hold}$ shows a further small elliptical feature at the location of highest optical density, which is consistent with the presence of a BEC. The appearance of a BEC is clearly shown in Fig. 2c, d, showing $y$-integrated density distributions. For short $t_{hold}$, only a broad, thermal distribution exists. However, for long $t_{hold}$, a bimodal distribution appears, the hallmark of a BEC.

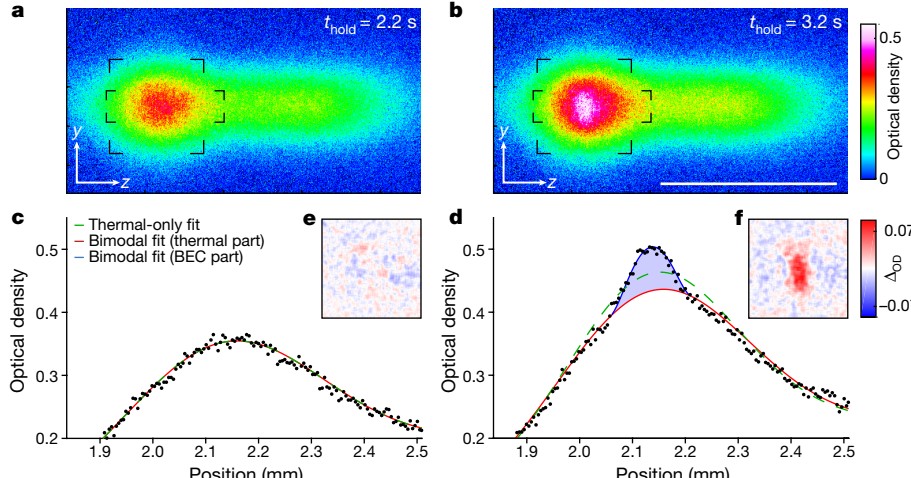

**Fig. 2 | Detection of the CW BEC. a**, **b**, Absorption images of the atomic cloud before and after condensation. The atoms are imaged after an 18-ms time-of-flight expansion. The scale bar denotes 1 mm. **c**, **d**, Optical density within the elongated rectangles marked by corners in **a** and **b**, averaged along *y*. Fitted profiles using a thermal-only distribution (green dashed line) or a bimodal distribution, consisting of a thermal (red line) and a Thomas–Fermi (blue line) component. The thermal-only fit fails to represent the condensed atoms in **d** (blue shaded area). **e**, **f**, Corner-marked square regions of absorption images **a** and **b** minus thermal parts of the bimodal fits, showing the CW BEC.

We further validate the existence of the BEC by fitting theoretical distributions to the absorption images in Fig. 2a, b. As shown in Fig. 2c, d, excellent agreement is found by combining a thermal distribution with a Thomas–Fermi distribution describing the BEC. At short hold times, we find that a thermal fit alone is sufficient to describe the data, whereas at longer times, the extra Thomas–Fermi component is required, indicating the presence of a BEC. To clearly visualize the BEC, we remove the thermal fit component from the data; see Fig. 2e, f.

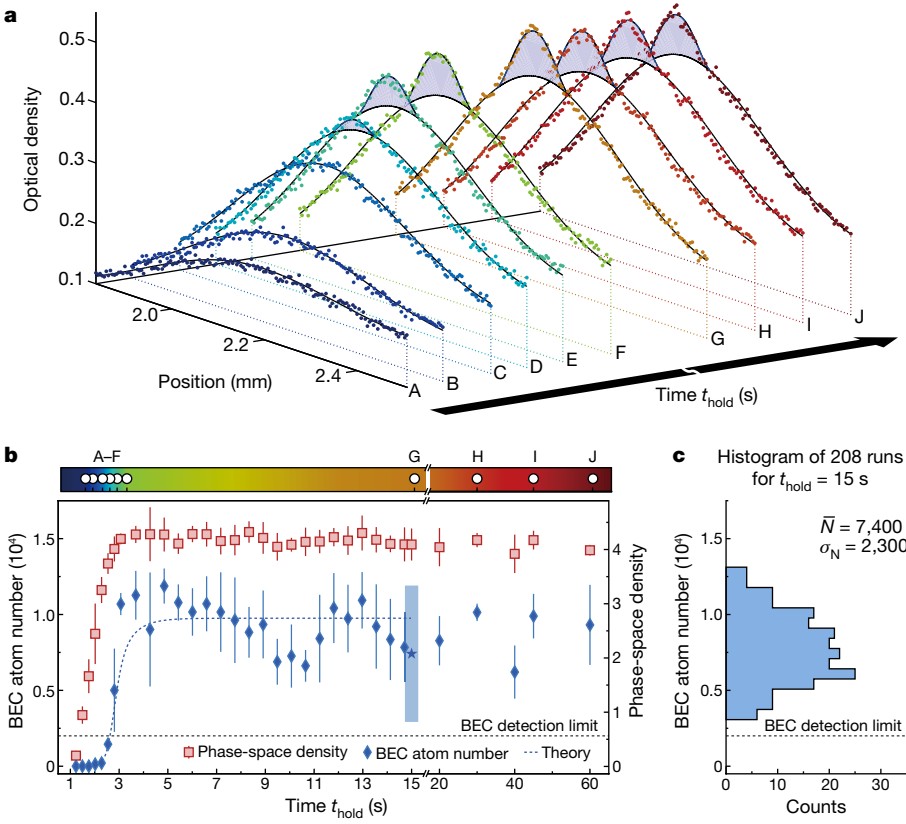

**Fig. 3 | Formation and stability of the CW BEC. a**, Profiles as in Fig. 2c, d for various hold times (marked in **b**) first during the formation of the BEC (A: 1.5 s, B: 1.8 s, C: 2.2 s, D: 2.5 s, E: 2.8 s, F: 3.2 s) and then during the steady state of the CW BEC (G: 15 s, H: 30 s, I: 45 s, J: 60 s). **b**, Evolution of the BEC atom number and the dimple atom phase-space density $\rho_D$ depending on hold time $t_{hold}$ after suddenly switching on all laser beams. The blue dashed line shows the result of the BEC evolution fitted to the data before 15 s using the rate-equation model (see Supplementary Information). The error bars show the standard deviation from binning about four measurements for each time. **c**, Histogram of the BEC atom number from 208 images for $t_{hold}$ = 15 s, long after the establishment of steady state (blue star in **b**). No points fall below our BEC detection limit of 2,000 atoms. The 95% confidence interval ($4\sigma_N$) calculated from this dataset is given in **b** at 15 s (blue rectangle).

The pronounced anisotropic shape of the BEC in Fig. 2f is consistent with the expansion of a BEC from the anisotropic dimple, whose trap frequency along the $y$ axis is approximately double that along $z$ (see Supplementary Information and Extended Data Fig. 6).

Once established, the BEC can be maintained in steady state indefinitely with gain balancing losses. As shown in Fig. 3, we study the formation transient and stability of the condensate by recording and analysing images for different $t_{hold}$. Figure 3a shows representative density profiles during the initial 5-s formation transient (A–F) and then in the presence of a stable BEC (G–J). Likewise, Fig. 3b shows the evolution and then stability of the BEC atom number and the peak phase-space density in the dimple, $\rho_D = N_D \left( \frac{\hbar^3 \omega_{Dx} \omega_{Dy} \omega_{Dz}}{k_B^3 T_D^3} \right)$, in which $N_D$ is the thermal atom number in the dimple and $\omega_{Di}/2\pi$ are the dimple trap frequencies. The steady-state BEC is observed over durations much longer than both the lifetime of a pure BEC (1.5–3 s) and the background-gas-limited lifetime (7 s) (see Methods).

Although we do not continuously monitor the CW BEC, its atom number fluctuations can be estimated from many independent observations. To study these fluctuations, we collected about 200 measurements for $t_{hold} = 15$ s, which is markedly longer than both the lifetimes in the system and the formation transient; see Fig. 3c. The average BEC atom number is $\overline{N} = 7{,}400(2{,}300)$, with none of the points falling below our BEC detection threshold of 2,000 atoms (see Methods).

Modelling the formation, growth and stabilization of the BEC provides valuable insights into this new driven-dissipative system. It also provides the gain and loss from the BEC, which are important for practical applications such as producing a CW atom laser[32] and improving matter-wave coherence. We explain the BEC dynamics by fitting a phenomenological rate-equation model to measured temperature and atom numbers. Our analysis covers the condensate formation and perturbations such as disrupting the reservoir loading (see Supplementary Information). From this model, we estimate a steady-state gain of $2.4(5) \times 10^5$ atoms s$^{-1}$ into the BEC, with representative fits shown in Fig. 3b and Extended Data Fig. 7. A substantial fraction of this gain could conceivably be translated into an outcoupled flux forming a CW atom laser. We also find that losses in the BEC at steady state are dominated by three-body recombinations with thermal atoms, owing to the gas density exceeding $5 \times 10^{14}$ atoms cm$^{-3}$. The presence of high, steady influx and loss makes our BEC a driven-dissipative system. We confirm this by showing that it is impossible to model the atoms in the trap as a closed system in thermal equilibrium (see Supplementary Information). Open driven-dissipative systems such as this one are thought to show rich non-equilibrium, many-body physics waiting to be explored, such as purity oscillations[49], behaviours described by new critical exponents[28] and unusual quantum phases, especially in lower dimensions[50].

## Discussion and conclusion

In summary, we have demonstrated continuous Bose–Einstein condensation. The resulting CW BEC can be sustained indefinitely using constant gain provided by a combination of Bose-stimulated scattering and atom refilling with high phase-space flux. Our work opens the door to continuous matter-wave devices. Moving forwards, many improvements are possible. In the near term, the purity of our BEC can be increased by enhancing the phase-space flux loading the dimple. A straightforward option to achieve this is to render the reservoir laser cooling uniform by using a magic-wavelength reservoir trap. Further options include lowering the reservoir temperature by Raman cooling[44] or by adding a continuously operating evaporation stage[36]. A CW BEC allows overcoming limits imposed on matter-wave coherence by the finite lifetime and atom number of a single condensate[51]. In practice, exceeding this limit will require extreme field stability, including external fields such as dipole trap laser fields and the condensate mean field.

For example, a coherence time exceeding 1 s requires an atom number stability on the order of 0.1%. Techniques such as feedback could be used to overcome such sources of noise[52,53] and could ultimately allow coherence approaching the standard quantum limit or beyond[51,54,55].

Our CW BEC is the matter-wave equivalent of a CW optical laser with fully reflective cavity mirrors. A tantalizing prospect is to add an output coupler to extract a propagating matter-wave. This could be implemented by coherently transferring atoms to an untrapped state and would bring the long-sought CW atom laser finally within reach[15,45]. This prospect is especially compelling because our CW BEC is made of strontium, the element used in some of today's best clocks[56] and the element of choice for future cutting-edge atom interferometers[20–24,57,58]. Our work could inspire a new class of such quantum sensors.

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

## Methods

### Creating an ultracold $^{84}$Sr beam

We use the experimental scheme developed in our previous work[7,39] to create an ultracold $^{84}$Sr beam propagating within a dipole trap guide. The scheme begins with strontium atoms emitted by an 850-K oven. They then travel through a succession of laser-cooling stages arranged along several connected vacuum chambers using first the $^1S_0$–$^1P_1$ and then the $^1S_0$–$^3P_1$ transitions. Using the 30-MHz-wide $^1S_0$–$^1P_1$ transition is necessary to efficiently slow and cool the fast atoms from the oven. However, this strong transition cannot be used in the last chamber in which the BEC is located, owing to the probable heating of the BEC from scattered near-resonant photons. Cooling using the narrow $^1S_0$–$^3P_1$ transition is, however, made possible in this last chamber thanks to the addition of a transparency beam (see below).

To form a guided beam, atoms arriving in the final vacuum chamber are first captured and cooled in a narrow-line MOT. They are then outcoupled into a long, horizontal dipole guide with a 92-µm waist. The $^{84}$Sr atoms propagate along the guide with a velocity $v_G = 8.8(8)$ cm s$^{-1}$, a Gaussian velocity spread $\Delta v_G = 5.3(2)$ cm s$^{-1}$ and a flux $\Phi_G = 8.6(1.0) \times 10^6$ atoms s$^{-1}$.

### Making the reservoir and dimple traps

The 11.5-µK-deep reservoir is produced by a right circularly polarized 1,070-nm laser beam propagating in the $z$ direction. It uses 540 mW of power focused to an elliptical spot with waists of $w_y = 14.5$ µm vertically and $w_x = 110$ µm horizontally. The guided atomic beam and the reservoir intersect with a horizontal angle of 6° about 1 mm from the reservoir centre and 37 mm from the MOT quadrupole centre. The reservoir beam crosses approximately 45(10) µm below the guide beam and descends with a vertical tilt of around 1.2(1)° as it separates from the guide beam. A secondary 250-mW beam of waist 175(25) µm runs parallel to the guide and points at the reservoir region. The fine adjustment of these beams is used to optimize the flow of atoms from the guide to the reservoir.

The dimple region has a 7 µK deeper potential located at the centre of the reservoir. This is mainly produced by a vertically propagating 1,070-nm 'dimple beam', although 1 µK is due to the vertically propagating transparency beam. The dimple beam uses 130 mW of power linearly polarized along the $z$ axis with a 27-µm waist in the plane of the reservoir. The dimple trap frequencies are $(\omega_{Dx}, \omega_{Dy}, \omega_{Dz}) = 2\pi \times (330, 740, 315)$ Hz, whereas the reservoir beam alone produces a trap with frequencies $(\omega_{Rx}, \omega_{Ry}, \omega_{Rz}) = 2\pi \times (95, 740, 15)$ Hz.

### Zeeman slower on the $^1S_0$–$^3P_1$ transition

To load the guided atomic beam into the reservoir, it must first be slowed and pushed into the reservoir. To perform this task, we implement a Zeeman slower using the $^1S_0$–$^3P_1$ transition starting around 3 mm before the guide–reservoir intersection. The slower makes use of the quadrupole magnetic field of the narrow-line MOT to provide a magnetic gradient along the axis of the guide. The quadrupole field of the MOT has gradients of −0.55, 0.32 and 0.23 G cm$^{-1}$ in the $x$, $y$ and $z$ directions respectively. The slower is displaced by 37 mm along the $z$ axis with respect to the quadrupole centre, resulting in a magnetic field offset of 0.85 G. The slower uses a counterpropagating 200-µm-waist laser beam that crosses the guide at a shallow horizontal angle of 4°. We modulate the laser frequency to broaden its effective linewidth to 50 kHz. This makes the slowing robust to potential fluctuations in the effective detuning (see Extended Data Table 1). The light intensity corresponds to 2.2 $I_{sat}$ when not frequency-broadened, in which $I_{sat} \approx 3$ µW cm$^{-2}$ is the saturation intensity of the transition. We choose the laser detuning to match the Zeeman shift of the $^3P_1|J'=1, m'_J=-1\rangle$ state at the intersection between the guide and the reservoir. This way, atoms reach zero axial velocity at the intersection before being pushed back and into the reservoir.

### Loading the reservoir

Because the reservoir is a conservative trap, efficiently loading atoms from the guide requires a dissipative mechanism. This is provided in two ways by laser cooling on the $^1S_0$–$^3P_1$ transition. The first is a 'counter Zeeman slower' beam propagating approximately along the $z$ axis opposing the Zeeman slower beam. This beam addresses the $^3P_1|J'=1, m'_J=-1\rangle$ state with a peak intensity of about 8 $I_{sat}$ and has a waist of 150 µm. Making use of this magnetic transition, we choose the light detuning such as to address the atoms near the guide–reservoir intersection and thus compensate the backwards acceleration of the Zeeman slower beam. This allows atoms to gradually diffuse towards the reservoir centre, in which collisions and the second laser-cooling mechanism will further lower their temperature.

The second cooling mechanism consists of a molasses on the radial axes $(x, y)$ addressing the magnetically insensitive π transition. Using a magnetically insensitive transition avoids affecting cooling by the spatial inhomogeneities in the effective detuning owing to magnetic field variation across the extent of the laser-cooled cloud. Another cause of spatial inhomogeneities, which does affect the molasses cooling efficiency, is the differential light shift induced by the reservoir trap. This shift is around +55 kHz, many times larger than the linewidth of the transition. The optimal molasses cooling frequency is found to be 42 kHz higher than the unperturbed transition. This partially accommodates for the differential light shifts and preferentially cools atoms located near the bottom of the reservoir. To reach the lowest temperature and enable condensation in the dimple, we also apply a very low total light intensity of 0.4 $I_{sat}$. With this choice of detuning and intensity, some of the incoming atoms reach the reservoir centre, in which they are radially cooled to $T_{Rr} = 0.85(7)$ µK. Other atoms might be heated out of the 9-µK-evaporation-threshold trap by the blue-detuned light in the outer trap region.

### Minimizing heating and loss in the reservoir

The atoms in the reservoir have a lifetime of 7 s, limited by collisions with the background gas of the vacuum chamber. However, these losses can be overwhelmed by optical effects such as photoassociation or heating by photon scattering. It is therefore critical to minimize the exposure of the reservoir to unnecessary light, and we address this point by implementing four techniques.

First, the 37-mm offset between the MOT and reservoir centres allows us to avoid any direct illumination from the $x$, $y$ MOT beams on the reservoir; see Extended Data Fig. 1. On the $z$ axis, the influence of the MOT beams is greatly reduced by using a 'dark cylinder', as described in ref. [39].

Second, we optimize the cooling spectrum and intensity of each laser-cooling beam entering the last vacuum chamber. By separately measuring their influence on the reservoir atom number, we optimize on a compromise between the lifetime of atoms and the loading flux. The results are illustrated in Extended Data Fig. 1 and Extended Data Table 1.

Third, we maximize the π polarization component of the molasses beams that illuminate both the guided beam and the reservoir, thus minimizing the effects of unwanted transitions. Unavoidably, beams along the $y$ axis possess admixtures of $\sigma^-$ and $\sigma^+$ owing to the orientation of the local magnetic field.

Finally, we purify the spectrum of the light used to address the $^1S_0$–$^3P_1$ cooling transition. Our cooling light is produced by several injection-locked diode lasers beginning from a single external-cavity diode laser (ECDL). We reduce the linewidth of this ECDL to 2 kHz by locking it onto a cavity with a finesse of approximately 15,000, whose spectrum has a full width at half maximum of about 100 kHz. By using the light transmitted through this cavity to injection lock a second diode laser, we can filter out the amplified spontaneous emission of the ECDL and servo bumps. This filtering is critical to increase the lifetime of the atoms inside the dimple by reducing resonant-photon scattering.

Without the dimple and transparency beams, individual laser-cooling beams reduce the lifetime of atoms in the reservoir to no shorter than about 1.5 s. With the dimple, transparency and all laser-cooling beams on, atoms in the reservoir have a 1/e lifetime of 420(100) ms, as determined from the fits shown in Extended Data Fig. 2.

## Transparency beam

To minimize the destructive effects of resonant light on the BEC and atoms within the dimple, we render this region locally transparent to light on the $^1S_0$–$^3P_1$ cooling transition. By coupling light to the $^3P_1$–$^3S_1$ transition, we induce a light shift on the $^3P_1$ state, as illustrated in Extended Data Fig. 3a, b. Owing to the extreme sensitivity of the BEC to photon scattering, all sub-levels of the $^3P_1$ state must be shifted markedly. This requires using at least two of the three transition types ($\sigma^\pm$, $\pi$) in this $J = 1$–$J' = 1$ structure. However, when polarizations at the same frequency are combined, quantum interference between sub-levels always produces a dark state in the dressed $^3P_1$ manifold. In this case, the energy of this dark state can only be shifted between $\pm\Delta_{Zeeman}$, in which $\Delta_{Zeeman}$ is the Zeeman shift of the $^3P_1 m'_J = 1$ state. This corresponds to $\Delta_{Zeeman} = 1.78$ MHz at the dimple location, giving a light shift that is insufficient to protect the BEC. Thus it is necessary to use different frequencies for the different polarization components of the transparency beam, as illustrated in Extended Data Fig. 3c.

The transparency beam is implemented by a single beam propagating vertically and focused on the dimple location with a 23-μm waist. This geometry aims to minimize the overlap of the transparency beam with the reservoir volume. In this way, we protect atoms at the dimple location without affecting the laser cooling taking place in the surrounding reservoir. This is necessary to maintain the high phase-space flux of the reservoir. The transparency laser light is blue detuned by 33 GHz from the 3.8-MHz-wide $^3P_1$–$^3S_1$ transition at 688 nm. This detuning is chosen to be as large as possible while still enabling sufficient light shift with the available laser power. The light contains two frequency components: 7 mW of right-hand circularly polarized light and 3 mW of left-hand circularly polarized light, separated by 1.4 GHz. The relative detuning is chosen to be large enough to avoid dark states while remaining experimentally easy to implement. It is small compared with the absolute detuning to obtain similarly good protection by each component. The relative intensity is chosen to shift all $^3P_1$ states by a similar magnitude. The magnetic field at the dimple location lies in the $(y, z)$ plane and has an angle of 60° with respect to the vertical $y$ axis along which the transparency beam propagates. This leads to a distribution of the light intensity onto the transitions $\{\sigma^+, \sigma^-, \pi\}$ of $\{1, 9, 6\}$ for the left-hand and $\{9, 1, 6\}$ for the right-hand circular polarization.

The light is produced from a single ECDL, frequency shifted by acousto-optic modulators and amplified by several injection-locked laser diodes and a tapered amplifier. Because the $^1S_0$–$^3P_1$ and $^3P_1$–$^3S_1$ lines are less than 1.5 nm apart, it is crucial to filter the light to prevent amplified spontaneous emission from introducing resonant scattering on the $^1S_0$–$^3P_1$ transition. This filtering is performed by a succession of three dispersive prisms (Thorlabs PS853 N-SF11 equilateral prisms), followed by a 2.5-m (right-hand circular) or 3.9-m (left-hand circular) propagation distance before aperturing and injection into the final optical fibre.

## Characterizing the transparency beam protection

The transparency-beam-induced light shifts on the $^1S_0$–$^3P_1$ transition were measured spectroscopically by probing the absorption of $^{88}$Sr samples loaded into the dimple. $^{88}$Sr is used instead of $^{84}$Sr because the higher natural abundance improves signal without affecting the induced light shifts. Spectra are recorded for various transparency beam laser intensities at the magnetic field used for the CW BEC experiments. The results are shown in Extended Data Fig. 3c for one and then two polarization components.

The observed light shifts are consistent with the calculated dressed states for the six coupled sub-levels of the $^3P_1$ and $^3S_1$ states. This is evaluated by solving the Schrödinger equation in the rotating frame of the light field for a transparency beam consisting of a single-frequency, right-hand circular laser beam in the presence of the measured external magnetic field. The theoretical results are given in Extended Data Fig. 3c (solid lines, left side), with no adjustable parameters. We find a reasonable agreement with the observed shifts and reproduce the expected saturation of the light shift owing to the presence of a dark state. An optimized fit can be obtained with a slightly higher intensity corresponding to a waist of 21 μm instead of 23 μm, and a slightly modified polarization distribution. In this fitted polarization distribution, the contribution of the weakest component, $\sigma^-$, is enhanced by a factor of roughly 2.5. Both differences can be explained by effects from the vacuum chamber viewports and dielectric mirrors.

When the left-hand circular polarization component of the transparency beam is added, we observe in Extended Data Fig. 3c (right side) that the 'dark' state shifts linearly away. In this manner, all sub-levels of $^3P_1$ can be shifted by more than 4 MHz, more than 500 times the linewidth of the laser-cooling transition. For comparison, the light shift on the $^1S_0$ ground state from the transparency beam is 20 kHz, and at most 380 kHz by all trapping beams, about one order of magnitude smaller than the shift on $^3P_1$ states from the transparency beam.

We demonstrate the protection achieved by the transparency beam in two ways. First, we measure the lifetime of a pure BEC inside the dimple in the presence of all light and magnetic fields used in the CW BEC experiments. This pure BEC is produced beforehand using time-sequential cooling stages. Once the pure BEC is produced, we apply the same conditions as used for the CW BEC, except that the light addressing the $^1S_0$–$^1P_1$ transition is off, to prevent new atoms from arriving. Without the transparency beam, the 1/e lifetime of a pure BEC in the dimple cannot even reach 40 ms, whereas with the transparency beam, it exceeds 1.5 s.

Second, we show the influence of the transparency beam on the existence of a CW BEC. Beginning with the same configuration as the CW BEC but without the transparency beam, steady state is established after a few seconds, with no BEC formed. We then suddenly switch the transparency beam on and observe the evolution of the sample as shown in Extended Data Fig. 4. Although the reservoir sample seems unaffected, the dimple atom number increases by a factor of 6.4(1.8), indicating fewer losses. At the same time, the sample (partially) thermalizes and a BEC appears after about 1 s. No BEC is formed if only one transparency beam frequency component is present or only one-third of the nominal transparency beam power is applied. This demonstrates the critical importance of the transparency beam.

## Characterizing the BEC and thermal cloud

To characterize the CW BEC and surrounding thermal cloud, we switch all traps and beams off and perform absorption imaging. Fitting the distributions of the expanding clouds allows us to estimate atom numbers and temperatures throughout the system, as well as the number of condensed atoms, all from a single image.

We begin with absorption images typically recorded after an 18-ms time-of-flight expansion. The observed 2D density distribution can be fitted by an ensemble of four thermal components plus an extra Thomas–Fermi distribution when a BEC is present. Three independent 2D Gaussian functions represent atoms originating from the dimple, the reservoir and the crossing between the guide and the reservoir. Atoms originating from the guide are represented along the guide's axis by a sigmoid that tapers off owing to the effect of the Zeeman slower and in the radial direction by a Gaussian profile. Examples are shown in Extended Data Fig. 5.

We found this fit function with 18 free parameters to be the simplest and most meaningful one capable of representing the data. By combining knowledge of their distinct locations and/or momentum spreads,

we can determine individually the populations and their characteristics. We find that the uncertainty in the fitted parameters is mostly unimportant compared with shot-to-shot variations in the data. An exception is distinguishing the population in the reservoir from that in the guide–reservoir crossing region, in which there is some ambiguity, resulting in higher uncertainties. In both the main text and Methods, the error bars indicate the standard deviation $\sigma$ calculated from several images. Although it is possible to estimate the temperatures in the $y$ axis from a single fitted image, the initial cloud sizes in the $z$ direction are large compared with the ballistic expansion. Thus we use a set of measurements with varying times of flight to estimate $z$-axis temperatures.

When a BEC is present, it is necessary to add a Thomas–Fermi profile to the previously discussed fit function. The only other free parameter used in the fit is the number of atoms in the BEC. We assume that the BEC position is the same as that of the non-condensed atoms in the dimple and we calculate the radii of the BEC from the BEC atom number, the $s$-wave scattering length, the trap frequencies in the dimple and the expansion time[59]. These frequencies are calculated from the knowledge of the waists of each relevant beam and of the powers used. The waists are either directly measured or extracted from observations of dipole oscillation frequencies of a pure BEC in the trap for several beam powers.

Adding an extra fitting parameter can lead to overfitting. To rigorously determine whether including this Thomas–Fermi distribution provides a significantly better fit of the data, we use a statistical $F$-test. This allows us to determine a BEC atom number threshold above which the fit is statistically better than that without the Thomas–Fermi distribution. For this $F$-test, we isolate a region of interest (ROI) in the image containing both thermal and BEC atoms. We then calculate the value $F = \frac{(\mathrm{RRS}_1 - \mathrm{RSS}_2)}{p_2 - p_1} / \frac{\mathrm{RSS}_2}{n - p_2}$, in which $\mathrm{RRS}_i$ is the residual sum of squares over the ROI for model $i$ with $p_i$ parameters and $n$ is the number of pixels of the ROI. The fit including the Thomas–Fermi distribution is significantly better than that without only if $F$ is higher than the critical value of an $F$-distribution with $(p_2 - p_1, n - p_2)$ degrees of freedom, with a desired confidence probability. By applying this test to the data of Fig. 3, we find that the BEC model fits better, with a confidence greater than 99.5%, when the BEC atom number exceeds 2,000. This sets our detection limit, above which we are confident a BEC exists. Notably, this limit is lower than the BEC atom number, corresponding to a $-2\sigma_N$ shot-to-shot fluctuation. This shows that, at all times after steady state is reached, a BEC exists.

## Data availability

Raw data and analysis materials used in this research can be found at https://doi.org/10.21942/uva.16610143.v1.

59. Castin, Y. & Dum, R. Bose-Einstein condensates in time dependent traps. *Phys. Rev. Lett.* **77**, 5315–5319 (1996).

**Acknowledgements** We thank F. Famá, S. Pyatchenkov, J. Samland, S. Zhou and the workshop of FNWI, especially J. Kalwij, T. van Klingeren and S. Koot, for technical assistance. We thank V. Barbé, C. Coulais, S. Diehl, K. van Druten, D. Guéry-Odelin, W. von Klitzing, B. van Linden van den Heuvell, R. Spreeuw, J. Tan, P. Thekkeppat and J. Walraven for comments on the manuscript. We thank the NWO for funding through Vici grant no. 680-47-619 and NWA Startimpuls 2 grant NWA.QUANTUMNANO.2019.002. We thank the European Research Council (ERC) for funding under project no. 615117 QuantStro. This project has received funding from the European Union's Horizon 2020 research and innovation programme under grant agreement no. 820404 (iqClock project). This work was financially supported by the Dutch Ministry of Economic Affairs and Climate Policy (EZK), as part of the Quantum Delta NL programme. B.P. thanks the NWO for funding through Veni grant no. 680-47-438 and C.-C.C. thanks Taiwan's Ministry of Education for MOE Technologies Incubation Scholarship no. 60010200068.

**Author contributions** C.-C.C. and S.B. built the apparatus. C.-C.C., R.G.E. and S.B. performed the investigation and data collection. C.-C.C., B.P. and S.B. analysed the data. J.M. developed the theoretical model. B.P., S.B. and F.S. supervised the project. F.S. acquired funding. All authors contributed to the manuscript.

**Competing interests** The authors declare no competing interests.

**Additional information**
**Correspondence and requests for materials** should be addressed to Florian Schreck.

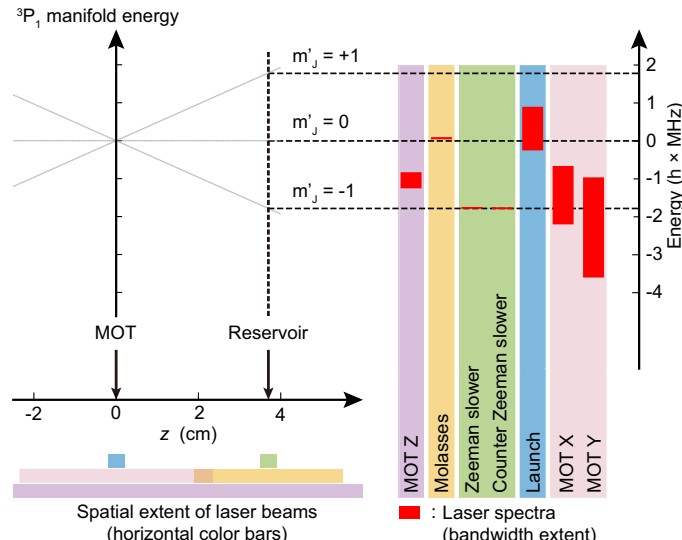

**Extended Data Fig. 1 | Spectra of narrow-linewidth cooling lasers and their spatial extent.** The right side represents the spectra of cooling lasers addressing the $^1S_0-^3P_1$ transition (vertical red bars) with respect to the (relative) energy of the states in the $^3P_1$ manifold, shown on the left side. The energies of these $m'_J$ states are given depending on the location along the $z$ axis, and the horizontal black dashed lines represent their respective Zeeman shifts when atoms are located inside the reservoir. The horizontal colour bars at the bottom left show the location and spatial extent of each laser beam; see also Extended Data Table 1 for detailed beam parameters.

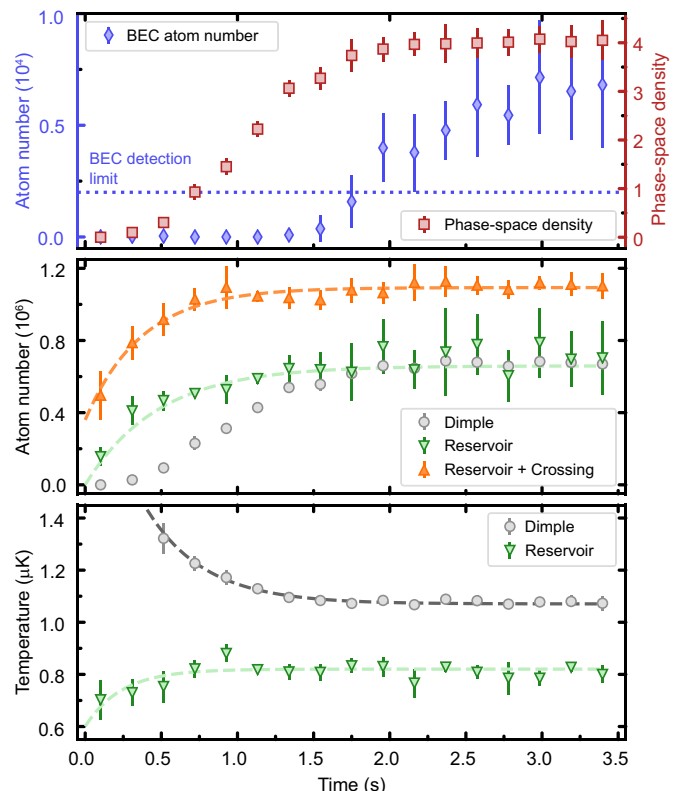

**Extended Data Fig. 2 | Loading of the reservoir and dimple at constant flux.**
We achieve a constant flux $\Phi_G$ in the guide by switching the experiment on for 10 s without the Zeeman slower beam, until reaching a steady flow. We then switch this beam on at time $t = 0$. We show the BEC atom number and the phase-space density $\rho_D$ in the dimple (top). The blue dotted line indicates our BEC detection limit in terms of condensed atom number. We show the dimple, reservoir and 'reservoir + crossing' atom number (middle) and the temperature $T_D$ in the dimple and the temperature $T_{Ry}$ in the reservoir along the vertical axis (bottom). The dashed lines are the results from fits with exponential growth or decay, giving access to the (constant) fluxes, one-body loss rate parameters and thermalization times (see Supplementary Information). The error bars represent one standard deviation $\sigma$ from binning, on average, six data points.

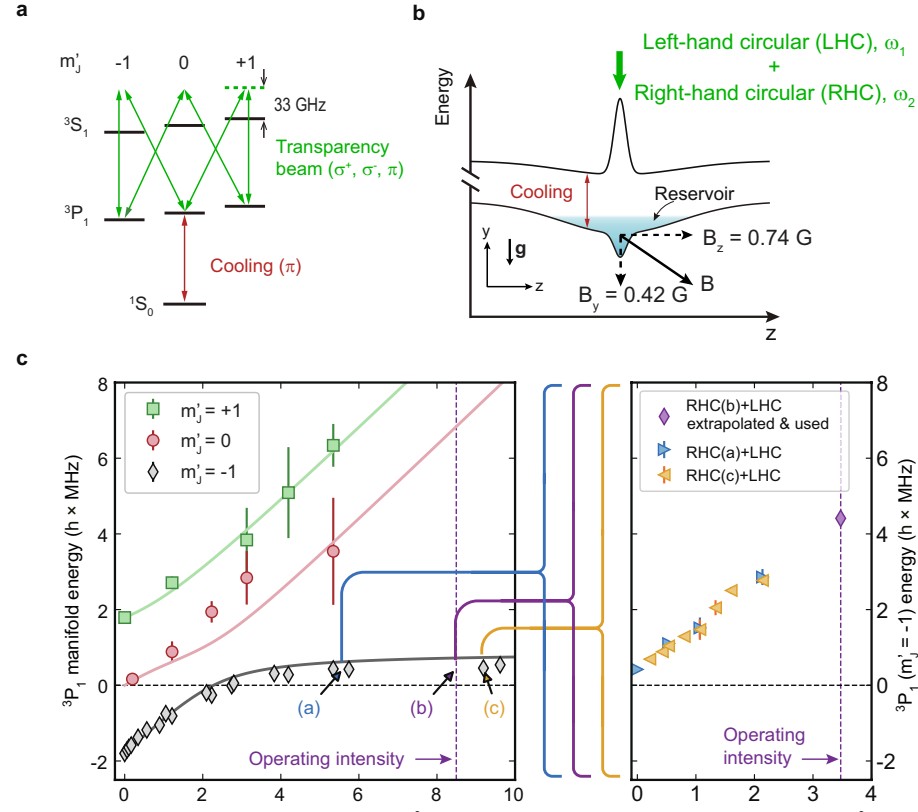

**Extended Data Fig. 3 | Light shift from the transparency beam. a**, Level scheme showing laser-cooling and transparency transitions. **b**, Schematic of the potential energy landscape of reservoir and dimple for the $^1S_0$ and $^3P_1$ states. Atoms are rendered insensitive to the laser-cooling light by a single vertical 'transparency' laser beam (green arrow) containing two frequency components, one for each circular polarization. **c**, Transition energies to the three $m'_J = 0, \pm1$ Zeeman sub-levels of the $^3P_1$ manifold, referenced to the transition at zero electric and magnetic field (black dashed line). The energy shifts are shown for a single right-hand circular (RHC) polarization (left) and with the addition of the left-hand circular (LHC) component (right). We show the solutions (solid lines) of the Schrödinger equation for the $^3P_1$ manifold coupled by a light field with single frequency component and RHC polarization. In this case, at high laser intensities, the energy of the state originating from $m'_J = -1$ saturates, corresponding to the presence of a dark state. The vertical purple dashed lines show the operating intensities of the LHC and RHC light fields used in the CW BEC experiment, and the purple diamond is extrapolated from the data. The error bars indicate estimates of the ranges in which the light-shifted spectral lines lie.

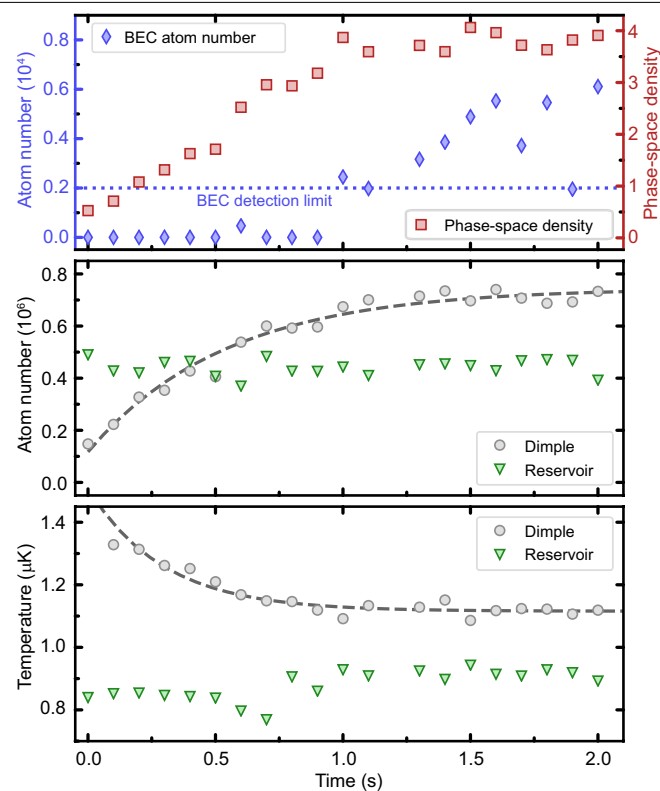

**Extended Data Fig. 4 | Influence of the transparency beam.** We let the
experiment reach a steady state with the transparency beam off. At $t = 0$, we
switch the beam on and observe the system's evolution. We show the BEC atom
number and the phase-space density $\rho_D$ in the dimple (top). The blue dotted line
indicates our BEC detection limit in terms of condensed atom number. We
show the dimple and reservoir atom number (middle) and the temperature $T_D$
in the dimple and the temperature $T_{Ry}$ in the reservoir along the vertical axis
(bottom). Both atom number and temperature in the reservoir remain constant
while the dimple loads further atoms, indicating lower losses thanks to the
protecting effect of the transparency beam.

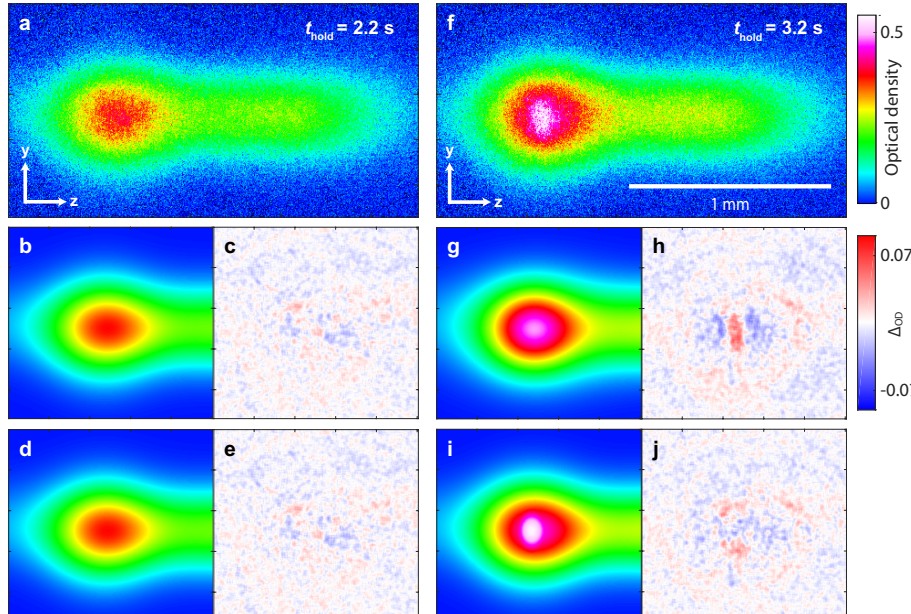

**Extended Data Fig. 5 | Fitting a CW BEC.** We show absorption pictures and their respective fits for two hold times, before (left, $t_{hold}$ = 2.2 s) and after (right, $t_{hold}$ = 3.2 s) the formation of the BEC. In the top row, we show absorption pictures **a**, **f** taken after an 18-ms time-of-flight expansion. In the middle row, we show results of fits **b**, **g** to these pictures and the fit residuals **c**, **h**. The fits use a 2D density distribution fit function accounting only for a thermal cloud.

By contrast, the bottom row shows both fits **d**, **i** and residuals **e**, **j** with a 2D density distribution fit function including a Thomas–Fermi distribution describing a BEC, in addition to the thermal distribution. In the presence of a BEC, the residual of the thermal-only fit **h** clearly shows a discrepancy at the BEC location, whereas the residual **j** demonstrates that the fit accounts for the BEC.

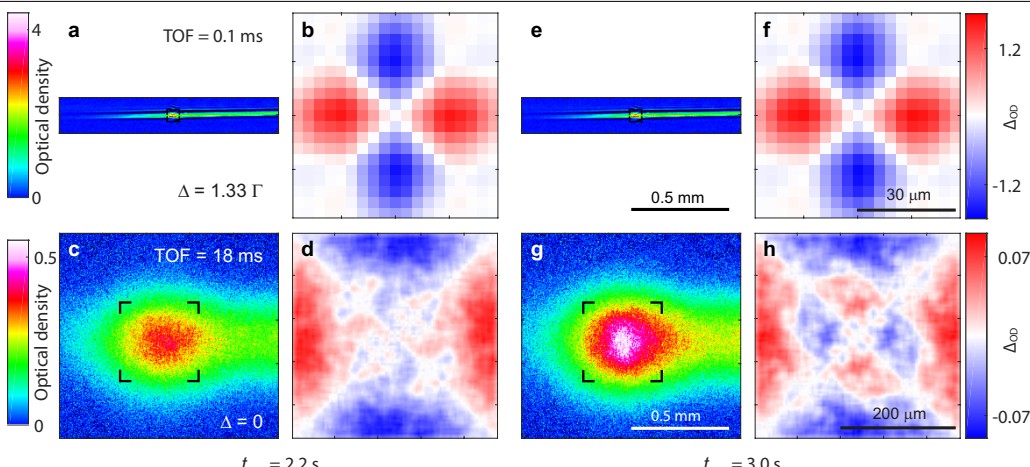

**Extended Data Fig. 6 | BEC anisotropy after time of flight.** Absorption images after short (0.1 ms) (**a**, **e**) and long (18 ms) (**c**, **g**) free-expansion time of flight (TOF), for short ($t_{hold}$ = 2.2 s, **a**, **c**) and long ($t_{hold}$ = 3.0 s, **e**, **g**) trap loading times. Pictures **a** and **e** were imaged with a detuning of 1.33$\Gamma$ to avoid saturation. The regions of interest (corner-marked squares) centred around the density maximums are analysed in panels **b**, **d**, **f** and **h**, which show the transpose anisotropy of the density distribution. This representation produces a cloverleaf pattern when the atomic cloud is anisotropic (see Supplementary Information). For short $t_{hold}$ (left), the cloverleaf pattern, which appears because of how the trap geometry initially shapes the thermal cloud, keeps a constant sign and diminishes during the expansion of a thermalized gas sample. For long $t_{hold}$ (right), we observe at long expansion time **h** another smaller cloverleaf pattern of opposite sign, which is indicative of the presence of a BEC.

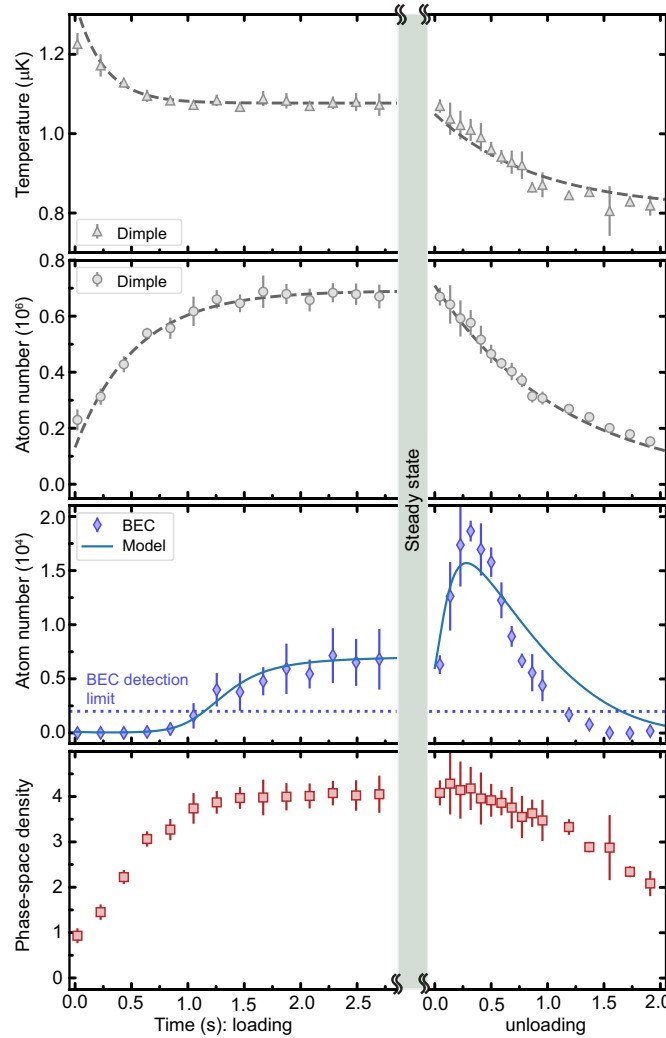

**Extended Data Fig. 7 | Modelling the onset and disappearance of the BEC.**
Evolution of the gas in the dimple. Left, BEC onset when loading reservoir and
dimple at constant flux $\Phi_G$ in the guide, started by Zeeman slower switch-on at
$t = -0.7$ s (same data as in Extended Data Fig. 2 but choosing a different origin
for the time axis). Right, BEC disappearance after setting $\Phi_G$ to zero by
switching the Zeeman slower off. The first two rows show the temperature and
atom number of the thermal atoms in the dimple. The last two rows show the
BEC atom number and the estimated phase-space density $\rho_D$ in the dimple. The
solid blue line is the result of the fit of the BEC atom number with the model of
Equation (10) of the Supplementary Information. The grey dashed lines are the
results of fits with exponential growth/decay functions, which are used as
input for Equation (10). The error bars represent one standard deviation $\sigma$ from
binning about four data points. For the BEC atom number, owing to the small
number of data points, the error bars can be underestimated compared with
the more reliable characterization at steady state of $\sigma_N = 2{,}300$ provided in the
main text. The phase-space density is estimated from measurements of the
atom number and temperature in the dimple in the same way as for Fig. 3.

**Extended Data Table 1 | Properties of laser beams addressing the narrow-linewidth $^1S_0$–$^3P_1$ transition**

| Beam name | Detuning (MHz) | Total power (µW) | $1/e^2$ radius (mm) | Comments |
|---|---|---|---|---|
| MOT X | −0.66 : 0.015 : −2.2 | $1.2 \times 10^3$ | 23.5 | two counter-propagating beams |
| MOT Y | −0.96 : 0.02 : −3.6 | $11.3 \times 10^3$ | 34 | single beam, upward propagating |
| MOT Z | −0.825 : 0.017 : −1.25 | 7 | 4 | two counter-propagating beams |
| Launch | +0.9 : 0.017 : −0.25 | $20 \times 10^{-3}$ | 0.25 | single beam |
| Zeeman slower | −1.74 : 0.017 : −1.79 | $4.5 \times 10^{-3}$ | 0.2 | single beam |
| Counter Zeeman slower | −1.77 : 0.017 : −1.79 | $10.5 \times 10^{-3}$ | 0.15 | single beam |
| Molasses X | +0.042 | 1.5 | 14.4 | two counter-propagating beams |
| Molasses Y(up) | +0.042 | 3.5 | 18 | single beam, upward propagating |
| Molasses Y(down) | +0.042 | $160 \times 10^{-3}$ | 19 | single beam, downward propagating |

In column 'Detuning', $\Delta_1$: δ: $\Delta_2$ refers to a comb of lines from detuning $\Delta_1$ to $\Delta_2$ with a spacing of δ, obtained by triangular frequency modulation.