## [Peer Review File · Nature]

Manuscript Title: Continuous Bose-Einstein condensation

Reviewer Comments & Author Rebuttals

Reviewer Reports on the Initial Version:

Referees' comments:

Referee #1 (Remarks to the Author):

The authors report the creation of a continuous Bose-condensed sample of strontium atoms, which constitutes a real milestone in AMO physics. The key ingredients that allow its realization, i.e. a continuous gain mechanism and a continuous supply of ultracold atoms, have been independently investigated both theoretically and experimentally in the last 20 years, and several intermediate steps have been achieved in the authors' group in the last few years. The achievement of this landmark objective is a major result, and importantly it paves the road for intriguing follow-up applications, one for all the CW atom lasers.

The authors give a precise and detailed summary of the key advances that one step at a time brought to their present result, each time underlying their meaning and limitations. That is the case of both the Bose-stimulated gain mechanisms and the continuous supply of cold atoms at high phase space density. The long quest undertaken by Schreck and co-workers to overtake the paradigm of time-sequential cooling stages towards ultracold atoms has proven successful, and the continuous BEC has now become a reality. How hot atoms evaporated in an oven continuously end up in the CW BEC is described in an exemplary way, with a wealth of quantitative details on the intermediate cooling steps - atomic flux, temperature, trap specifications. Particularly engaging is the section devoted to the demonstration that not only the BEC exists, but it is also continuous. The condensate lies within a thermal cloud, hence it is not trivial to demonstrate its presence; clear evidence is however reported, in terms of the observed bi-modal distribution and the anisotropic expansion dynamics of the atomic sample. That the BEC lasts indefinitely - i.e. the main claim of this article - is proven in several ways in order to dispel any possible doubts: the BEC existence is observed for intervals longer than the lifetime of a pure BEC and of the limitation imposed by the vacuum quality; the dynamics of the CW BEC is studied by interrupting and reactivating the BEC loading in key places of the space-sequential cooling, and a quantitative analysis of the losses mechanism. Subtle features, like the non-thermal equilibrium of the CW BEC, are analyzed in depth, with relevant measurements and specific modeling.

The manuscript is pleasant to read, always compelling, and what is most important, the scientific achievement is the commendable result of an incredible experimental effort. I certainly recommend the Editor to publish this manuscript in Nature, not last for its potential consequences in the field of atom-based quantum sensing.

The unique main remark I have concerns the possible implication of a CW BEC in atom

interferometry (AI); writing that the absence of a CW BEC prevents AI from reaching its full potential is probably an overstatement. AI does not intrinsically require a BEC source; most atom interferometers rely on single particle effects, and where quantum correlation between particles is exploited to boost sensitivity it is done in dilute samples, to avoid the hardly controllable interactions determined by the mean field potential (of course there are exceptions, but the applications listed in the manuscript are of this specific kind). The BEC is typically used for its high phase-space density, which brings to an optimal source after expansion and delta-kick collimation. To build their case on the potential on CW Bose-condensed source in atom interferometry, the authors should refer to several realizations of continuous atom interferometers, not mentioned in the actual version of the manuscript but highly relevant; they were introduced back in 1991 with Pritchard (PRL 66, 2693 - 1991), then widely adopted in gyroscopes, and are studied today to realize atom-aided inertial navigation (PRAppl 13, 044057 - 2020).

I also have some other comments and signal a few minor issues:

- A more recent article than [19] on the MAGIS-100 experiment is QST 6, 044003 (2021).
- In the conclusion section, to the list of proposals for future atom-interferometry space-based missions the authors could add AEDGE (EPJ Quantum Technol 7, 1 - 2020).
- Methods, lines 734-736 I do not see how the features of the transparency beam have been chosen (the two frequency components with different polarizations and intensity); the general aim seems to be to achieve the required light shift to protect the BEC at the center of the dimple.
- p2: What is meant by "irreversible loading" of the reservoir?
- do the authors have any clues of how stable is the BEC phase, specially in relation to the continuous filling of new atoms that could in some way steer the global coherence? Could this effect be characterized by the present setup?
- Suppl Mat: * line 30 "this" data should be better "these"
- * line 174 function ϵ should be singular * line 266 just for stylistic reasons, I would avoid using "disappearance" and "apparent" so close

Referee #2 (Remarks to the Author):

In this paper, the realization of continuous Bose-Einstein condensation is reported. The authors have reached the long-standing goal of maintaining a continuous gain process for the formation of the condensate and observed a steady-state Bose Einstein condensate.

This paper is very interesting and accomplished this goal with a combination of innovative techniques. It realizes the matter wave analog of a CW laser. The authors had to use a complex and impressive combination of laser-cooling techniques to continuously replenish a reservoir of ultracold atoms at high phase space densities without heating up the Bose-Einstein condensate by light scattering.

I think that the paper is in principle suitable for Nature, since it realizes a long-standing goal, and has the appeal of being the matter-wave analog of the CW laser. However, I feel the paper is incomplete in describing the potential of CW Bose-Einstein condensation and implies vast improvements over previous techniques which are misleading and overstated.

The introduction of the paper is almost bombastic and gives the impression that a continuous BEC will have the same impact as the CW laser and will vastly improve precision measurements and atom interferometry.

In analogy with lasers, I see two reasons why people want to use a continuous BEC as an atom source for experiments: (1) CW BEC can create a higher flux of condensed atoms. (2) A CW source can have a much longer coherence time than pulsed sources: Without the pulse-time limitation, they can approach the fundamental Schawlow-Townes limit. I don't think the current paper comes even close to deliver either on (1) or on (2).

Major activities in creating continuous atom lasers by beam slowing were motivated by using the full duty cycle of laser cooling. Most BEC experiments accumulate laser cooled atoms for only 10 % of the cycle time, e.g. 1 sec of laser cooling followed by 10 sec of evaporative cooling. With condensates of 10^7 atoms, this provides a flux of 10^6 condensed atoms per second, and if researchers were interested in improving those numbers (some of which were achieved 15 years ago), 10^7 should be in easy reach. Laser cooled atomic beams (often using Zeeman slowing) can have fluxes in excess of 10^{10} atoms per second. One goal, pursued over more than a decade, was to continuously cool such beams and possibly reach BEC fluxes of 10^7 or 10^8 atoms. In contrast, the current paper achieves a flux of about $3 \cdot 10^5$ atoms. For most precision experiments, where good statistics matters, CW condensates would be inferior. In my view, it is much easier and more efficient to optimize cooling efficiencies sequentially. So I don't expect that CW condensates will ever exceed the fluxes of pulsed condensates – the potential gain due to the duty-cycle is only a factor of ten or so, and easily lost by the complex requirements of shielding the ultracold atoms from resonant laser light.

Can a continuous BEC realize longer coherence times? In principle yes, but I think not in practice. There is no discussion about any limitations in the current paper. The authors give the misleading impression that they have developed a technique which will vastly improve the coherence time of matter-wave interferometers.

So what is currently possible? If you create a big BEC of 10^7 or 10^8 atoms, you could outcouple an atom laser beam of 10^6 atoms per second for tens of seconds. So what a CW atom laser has to beat is a coherence time of at least multiple seconds. I think this is far out of reach, or even impossible, and this is a big difference between optical lasers and atoms. In contrast to photons, atoms interact with each other and the environment.

To achieve a monochromatic atom laser beam with an energy resolution of 1 Hz would require the elimination of ALL possible energy shifts of this magnitude. If the BEC position in the gravitational field would fluctuate by 100 nm, it would change the kinetic energy of an atom laser output beam by 100 Hz. Collisional mean field shifts are typically in the kHz region. A stability of 1 Hz would require the control of the density of the condensate to within 0.1 %. In addition, electric and magnetic fields

have to be carefully controlled, but this should be possible.

I am not aware of any current experiment with ultracold atoms or Bose-Einstein condensates which is limited by the pulsed operation of the BEC source, since the coherence time is much more severely limited by other effects. Therefore, I don't expect that other researchers will immediately replicate the techniques described in the present paper. So I am challenging the authors to discuss possible applications within the next few years, where a CW condensate would provide an improvement in performance.

Despite my critical remarks, I think the authors have made an important advance which Nature could publish if the results are placed in the right context. For me, the present work is a proof-of-concept demonstration and will allow interesting studies of gain dynamics of matter waves and the characterization of non-equilibrium dynamics etc., but will not revolutionize atom interferometers.

The paper is well written and carefully edited and conveys the experimental techniques and analysis concisely. Important technical details are well presented in the supplemental information.

A few specific remarks:

- a. In my observation of the field, the goal of CW atom lasers was prominently pursued over decades with collisional cooling of cold atoms beams. This work should be more prominently cited since it shares the vision with the current work. This work was carried out by three groups (Paris, Utrecht, Michigan), but only the Paris group is cited. I know that the Utrecht work has not been published in regular journals, but it would be fair to include it (via conference proceeding of the Dutch annual meeting, or the 2011 Ph.D. thesis).
- b. The authors emphasize that the transparency beam increased the condensate lifetime from 40 m to 1.5 s, and present this as important for their achievements. However, under CW operation, the lifetime of the condensate due to three-body collisions is shortened to 30 ms. This suggests that a much less stringent requirement for the transparency beam would have been sufficient.
- c. Figure caption 2: Figures a and b show two rectangles. Which one is the region of interest for figures c and d?

Author Rebuttals to Initial Comments:

Response to referees

We would like to thank the referees for their thorough and positive reviews of this paper and their numerous helpful comments. We took all comments into account in the new version of the manuscript. In particular we have reframed our work, avoiding statements that can give the impression of widespread short term usefulness to practical applications. Instead we emphasize the strong points of our work also stressed by the referees.

Below we address each point brought up by the referees and indicate the corresponding changes made to the manuscript. We furthermore provide two manuscript versions that highlight text changes.

- A “redline + strikeout” version contains new text in red and deleted text as striken out red text. The line and reference numbers differ from the resubmitted version.
- A “redline” version contains new text in red, but no deleted text. This version has the same line and reference numbers as the resubmitted version.

Referees' comments (blue, italic, 10pt)

Authors response (black, roman, 11pt, with quotes in italic and modified or new text in bold)

Referee #1 (Remarks to the Author):

The authors report the creation of a continuous Bose-condensed sample of strontium atoms, which constitutes a real milestone in AMO physics. The key ingredients that allow its realization, i.e. a continuous gain mechanism and a continuous supply of ultracold atoms, have been independently investigated both theoretically and experimentally in the last 20 years, and several intermediate steps have been achieved in the authors' group in the last few years. The achievement of this landmark objective is a major result, and importantly it paves the road for intriguing follow-up applications, one for all the CW atom lasers.

The authors give a precise and detailed summary of the key advances that one step at a time brought to their present result, each time underlying their meaning and limitations. That is the case of both the Bose-stimulated gain mechanisms and the continuous supply of cold atoms at high phase space density. The long quest undertaken by Schreck and co-workers to overtake the paradigm of time-sequential cooling stages towards ultracold atoms has proven successful, and the continuous BEC has now become a reality. How hot atoms evaporated in an oven continuously end up in the CW BEC is described in an exemplary way, with a wealth of quantitative details on the intermediate cooling steps - atomic flux, temperature, trap specifications. Particularly engaging is the section devoted to the demonstration that not only the BEC exists, but it is also continuous. The condensate lies within a thermal cloud, hence it is not trivial to demonstrate its presence; clear evidence is however reported,

in terms of the observed bi-modal distribution and the anisotropic expansion dynamics of the atomic sample. That the BEC lasts indefinitely - i.e. the main claim of this article - is proven in several ways in order to dispel any possible doubts: the BEC existence is observed for intervals longer than the lifetime of a pure BEC and of the limitation imposed by the vacuum quality; the dynamics of the CW BEC is studied by interrupting and reactivating the BEC loading in key places of the space-sequential cooling, and a quantitative analysis of the losses mechanism. Subtle features, like the non-thermal equilibrium of the CW BEC, are analyzed in depth, with relevant measurements and specific modeling.

The manuscript is pleasant to read, always compelling, and what is most important, the scientific achievement is the commendable result of an incredible experimental effort. I certainly recommend the Editor to publish this manuscript in Nature, not last for its potential consequences in the field of atom-based quantum sensing.

1) Thank you.

The unique main remark I have concerns the possible implication of a CW BEC in atom interferometry (AI); writing that the absence of a CW BEC prevents AI from reaching its full potential is probably an overstatement. AI does not intrinsically require a BEC source; most atom interferometers rely on single particle effects, and where quantum correlation between particles is exploited to boost sensitivity it is done in dilute samples, to avoid the hardly controllable interactions determined by the mean field potential (of course there are exceptions, but the applications listed in the manuscript are of this specific kind). The BEC is typically used for its high phase-space density, which brings to an optimal source after expansion and delta-kick collimation. To build their case on the potential of CW Bose-condensed source in atom interferometry, the authors should refer to several realizations of continuous atom interferometers, not mentioned in the actual version of the manuscript but highly relevant; they were introduced back in 1991 with Pritchard (PRL 66, 2693 - 1991), then widely adopted in gyroscopes, and are studied today to realize atom-aided inertial navigation (PRAppl 13, 044057 - 2020).

- 2) We agree with the referee. We rewrote the paragraph discussing the relationship of atom interferometry and our work (paragraph 2). In particular,
- we deleted the statement “*However, no continuous matter-wave source with long-range coherence currently exists, which prevents atom interferometers from reaching their full potential*”,
 - we start the discussion of atom interferometry by stating that continuous operation has advantages and citing also the works quoted by the referee.

A more detailed discussion of our changes to the second paragraph is contained in our response to referee 2 below.

I also have some other comments and signal a few minor issues:

- A more recent article than [19] on the MAGIS-100 experiment is QST 6, 044003 (2021).

3) Thank you, we have replaced reference [19] (now [22]) with Ref. QST 6, 044003 (2021).

- In the conclusion section, to the list of proposals for future atom-interferometry space-based missions the authors could add AEDGE (EPJ Quantum Technol 7, 1 - 2020).

4) Thank you, we have followed the reviewer’s suggestion and included the reference AEDGE (EPJ Quantum Technol 7, 1 - 2020) as [58].

- Methods, lines 734-736 I do not see how the features of the transparency beam have been chosen (the two frequency components with different polarizations and intensity); the general aim seems to be to achieve the required light shift to protect the BEC at the center of the dimple.

5) The parameters were indeed chosen such that there is sufficient light shift to protect the BEC. We added the most important further considerations that led to the specific parameters.

We changed

“The transparency laser light is blue detuned by 33GHz from the 3P1-3S1 transition. It contains two frequency components, 7mW of right-hand circularly polarized light and 3mW of left-hand circularly polarized light separated by 1.4GHz”

to

*“The transparency laser light is blue detuned by 33GHz from the 3P1-3S1 transition. **This detuning is chosen as large as possible while still enabling sufficient light shift with the available laser power. The light** contains two frequency components, 7mW of right-hand circularly polarized light and 3mW of left-hand circularly polarized light separated by 1.4GHz. **The relative detuning is chosen large enough to avoid dark states while remaining experimentally easy to implement. It is small compared to the absolute detuning in order to obtain similarly good protection by each component. The relative intensity is chosen to shift all 3P1 states by a similar magnitude.**”*

- p2: What is meant by "irreversible loading" of the reservoir?

- 6) In the literature [Phys. Rev. A **57**, 2030 (1998), Nature Phys. **4**, 731 (2008), Phys. Rep. **529**, 265 (2013)] the notion of “irreversible loading” has frequently been described as a key requirement to achieve continuous condensation. This follows in analogy to the “irreversible pumping” requirement in optical lasers necessary to maintain inversion. In our case this refers to the cooling process, which increases the dimple phase-space density through a net transfer of atoms from the reservoir.

The choice of the term “irreversible” is perhaps confusing in the sense that individual atoms are exchanged between the reservoir and dimple and between the dimple and the BEC. We chose to continue to use the term in order to maintain consistency with the literature and with the analogies to optical lasers, but upon reflection this is probably more confusing than helpful.

We have therefore changed the text at line L128 from

*“This arrangement of traps and cooling beams leads to the **irreversible loading** of the reservoir with a flux...”*

to

*“This arrangement of traps and cooling beams leads to the **loading** of the reservoir with a flux...”*

- do the authors have any clues of how stable is the BEC phase, specially in relation to the continuous filling of new atoms that could in some way steer the global coherence? Could this effect be characterized by the present setup?

- 7) This is a very interesting question. While we are not able to measure the phase with our existing setup we are gradually working towards a system which we hope will provide a means to answer these questions similar to that published in Science **307**, 1945 (2005).

- Suppl Mat: * line 30 "this" data should be better "these"

8) Corrected, thank you.

* line 174 function s should be singular*

9) Corrected, thank you.

line 266 just for stylistic reasons, I would avoid using "disappearance" and "apparent" so close.

10) Thank you for the suggestion. In SM L266 we replaced

*"This corresponds to stopping the atomic flux into the reservoir and leads to the **disappearance of the BEC. It is apparent that the rate equation** Eq. (10) captures well the initial dynamics of the BEC atom number as well as the initial dynamics of the unloading stage"*

with

*"This corresponds to stopping the atomic flux into the reservoir and leads to the **disappearance of the BEC. The rate equation** Eq. (10) captures well the initial dynamics of the BEC atom number as well as the initial dynamics of the unloading stage."*

Referee #2 (Remarks to the Author):

In this paper, the realization of continuous Bose-Einstein condensation is reported. The authors have reached the long-standing goal of maintaining a continuous gain process for the formation of the condensate and observed a steady-state Bose Einstein condensate.

This paper is very interesting and accomplished this goal with a combination of innovative techniques. It realizes the matter wave analog of a CW laser. The authors had to use a complex and impressive combination of laser-cooling techniques to continuously replenish a reservoir of ultracold atoms at high phase space densities without heating up the Bose-Einstein condensate by light scattering.

I think that the paper is in principle suitable for Nature, since it realizes a long-standing goal, and has the appeal of being the matter-wave analog of the CW laser.

11) Thank you.

However, I feel the paper is incomplete in describing the potential of CW Bose-Einstein condensation and implies vast improvements over previous techniques which are misleading and overstated. The introduction of the paper is almost bombastic and gives the impression that a continuous BEC will have the same impact as the CW laser and will vastly improve precision measurements and atom interferometry.

In analogy with lasers, I see two reasons why people want to use a continuous BEC as an atom source for experiments: (1) CW BEC can create a higher flux of condensed atoms. (2) A CW source can have a much longer coherence time than pulsed sources: Without the pulse-time limitation, they can approach the fundamental Schalow-Townes limit. I don't think the current paper comes even close to deliver either on (1) or on (2).

Major activities in creating continuous atom lasers by beam slowing were motivated by using the full duty cycle of laser cooling. Most BEC experiments accumulate laser cooled atoms for only 10 % of the cycle time, e.g. 1 sec of laser cooling followed by 10 sec of evaporative cooling. With condensates of 10^7 atoms, this provides a flux of 10^6 condensed atoms per second, and if researchers were interested in improving those numbers (some of which were achieved 15 years ago), 10^7 should be in easy reach. Laser cooled atomic beams (often using Zeeman slowing) can have fluxes in excess of 10^{10} atoms per second. One goal, pursued over more than a decade, was to continuously cool such beams and possibly reach BEC fluxes of 10^7 or 10^8 atoms. In contrast, the current paper achieves a flux of about $3 \cdot 10^5$ atoms. For most precision experiments, where good statistics matters, CW condensates would be inferior. In my view, it is much easier and more efficient to optimize cooling efficiencies sequentially. So I don't expect that CW condensates will ever exceed the fluxes of pulsed condensates – the potential gain due to the duty-cycle is only a factor of ten or so, and easily lost by the complex requirements of shielding the ultracold atoms from resonant laser light.

Can a continuous BEC realize longer coherence times? In principle yes, but I think not in practice. There is no discussion about any limitations in the current paper. The authors give the misleading impression that they have developed a technique which will vastly improve the coherence time of matter-wave interferometers.

So what is currently possible? If you create a big BEC of 10^7 or 10^8 atoms, you could outcouple an atom laser beam of 10^6 atoms per second for tens of seconds. So what a CW atom laser has to beat is a coherence time of at least multiple seconds. I think this is far out of reach, or even impossible, and this is a big difference between optical lasers and atoms. In contrast to photons, atoms interact with each other and the environment.

To achieve a monochromatic atom laser beam with an energy resolution of 1 Hz would require the elimination of ALL possible energy shifts of this magnitude. If the BEC position in the gravitational field would fluctuate by 100 nm, it would change the kinetic energy of an atom laser output beam by 100 Hz. Collisional mean field shifts are typically in the kHz region. A stability of 1 Hz would require the control of the density of the condensate to within 0.1 %. In addition, electric and magnetic fields have to be carefully controlled, but this should be possible.

I am not aware of any current experiment with ultracold atoms or Bose-Einstein condensates which is limited by the pulsed operation of the BEC source, since the coherence time is much more severely limited by other effects. Therefore, I don't expect that other researchers will immediately replicate the techniques described in the present paper. So I am challenging the authors to discuss possible applications within the next few years, where a CW condensate would provide an improvement in performance.

Despite my critical remarks, I think the authors have made an important advance which Nature could publish if the results are placed in the right context. For me, the present work is a proof-of-concept demonstration and will allow interesting studies of gain dynamics of matter waves and the characterization of non-equilibrium dynamics etc., but will not revolutionize atom interferometers.

- 12) We take onboard the criticisms of the referee. We reworked the presentation and tone of the paper in the first two paragraphs and the outlook, trying to address the problems raised. Like the referee, we also see this work primarily as a proof-of-concept demonstration, which shows a way to break a long-standing barrier and an open challenge within the field. It will likely require years of work to translate our advance into gains for practical applications. We agree that in the short term the applications will be scientific, including the studies mentioned by the referee.

The main changes and their motivation in view of the referee's criticism are listed in the following (see the redline or redline+strikeout manuscript for context).

***** Introductory paragraph *****

12a) We changed

*"They are **essential** to quantum simulation [1] and sensing [2,3], for example underlying atom interferometers in space [4] and ambitious tests of Einstein's equivalence principle [5,6]"*

to

*"They are **important** to quantum simulation [1] and sensing [2,3], for example underlying atom interferometers in space [4] and ambitious tests of Einstein's equivalence principle [5,6]."*

because so far BECs are important for atom interferometry to reach long free flight times, but not generally essential.

12b) We deleted the sentence

"The key to dramatically increasing the bandwidth and precision of such matter-wave sensors lies in sustaining a coherent matter wave indefinitely"

and replaced it by

"A long-standing constraint for quantum gas devices has been the need to execute cooling stages time-sequentially, restricting these devices to pulsed operation."

This new sentence still gives an important motivation for our work, but avoids raising stretched expectations of practical usefulness for sensors.

(As a further, minor consequence of this change we deleted the sentence *"This advance overcomes a fundamental limitation of all atomic quantum gas experiments to date: the need to execute several cooling stages time-sequentially"* because that fact is now already mentioned earlier.)

12c) We deleted

"Continuous matter-wave amplification will make possible CW atom lasers, atomic counterparts of CW optical lasers that have become ubiquitous in technology and society"

because it can be read as implying that just because CW optical lasers are ubiquitous, CW atom laser will be as well. Instead we now simply state

"Our experiment is the matter wave analog of a cw optical laser with fully reflective cavity mirrors."

12d) We deleted

"The coherence of such atom lasers will no longer be fundamentally limited by the atom number in a BEC and can ultimately reach the standard quantum limit [9-11]"

because this statement can raise expectations that are too high.

This statement is still contained in the outlook (see below), but in a context that should make it clear that a lot of challenges still need to be solved and that approaching the coherence limit will take a long time to achieve.

12e) To make it even clearer that a lot of more work needs to be done before practical applications can arise from our work we changed

“Our development provides a new, hitherto missing piece of atom optics, enabling the construction of continuous coherent matter-wave devices”

to

*“**This proof of principle demonstration** provides a new, hitherto missing piece of atom optics, enabling the construction of continuous coherent matter-wave devices.”*

12f) We deleted the final sentence of the introductory paragraph

“From infrasound gravitational wave detectors [12,13] to optical clocks [14,15], the dramatic improvement in coherence, bandwidth and precision now within reach will be decisive in the creation of a new class of quantum sensors.”

because it can raise expectations that are too high, especially the expectation of usefulness of our work to sensing within a few years. The link between our work and quantum sensing is still contained in the second paragraph.

***** Second paragraph *****

12g) We rewrote the second paragraph to avoid the concerns of referee 2 and to address the comments of referee 1 listed under 2) above. The topic of the paragraph is still the relationship of our work to quantum sensing. We mention that continuous operation and BECs can be advantageous for sensors, and that in the long run cw atom laser beams could provide both advantages. As an aside we take up the argument of referee 2 that in the short term CW BECs are useful for scientific studies (as in the original submission this is also stated towards the end, in L253-258 (1st submission) and L244-248 (resubmission)).

The new version of the second paragraph is

*“**Continuous operation is advantageous for sensors as it eliminates deadtime and can offer higher bandwidths than pulsed operation [9–12]. Meanwhile, sensors employing Bose-Einstein condensates benefit from their high phase-space density and unique coherence properties [2–6,13]. Combining these advantages, a CW atom laser beam outcoupled from a CW condensate could be ideal for many quantum sensing applications [14–17]. In the long term CW atom lasers could benefit applications ranging from dark matter and dark-energy searches [18,19], gravitational-wave detection [20–24], tests of Einstein’s equivalence principle [5,6] to explorations in geodesy [25–27]. In the short term the CW BEC offers a platform to study quantum atom optics and novel quantum phenomena arising in driven-dissipative quantum gases [28].**”*

***** Outlook *****

12h) We modified the outlook to make it clear that many effects influence the phase of a BEC and a lot of work needs to be done to approach the fundamental limit imposed by time-sequential BEC production. We hope that our more careful wording avoids giving the

impression that we claim that a long coherence time is reachable in the short term. Therefore we believe a detailed discussion of effects that limit the coherence is not needed in this manuscript.

We deleted the even longer term outlook on squeezed states.

We changed the wording to avoid giving the impression that short term practical applications of our work are within sight. In the second to last sentence we updated references as suggested by referee 1, see 3) and 4) above.

Specifically, we changed the old L264-L291

“Our work paves the way for continuous matter-wave devices, where matterwave coherence is no longer limited by the atom number and lifetime of a single condensate [10]. In the near future, BEC purity and matter-wave coherence can be improved by enhancing the phase-space flux loading the dimple. [...unchanged text...] Real-time non-destructive detection and feedback [45] can be used to stabilize the CW BEC atom number to a stability approaching the shot-noise level [46] and even beyond, leading to squeezed states [47] for measurements beyond the standard quantum limit [10]. [...unchanged text...] [12,13,19-21,50]. Our will inspire a new class of such quantum sensors.”

to the new L255-283

*“Our work **opens the door to continuous matter-wave devices. Moving forwards many improvements are possible.** In the near **term, the purity of our BEC can be increased** by enhancing the phase-space flux loading the dimple. [...unchanged text...] **Techniques like feedback could be used to overcome many sources of noise and stabilize the CW BEC atom number and phase [51,52]. In the long term, matter-wave coherence will no longer be fundamentally limited by the atom number and lifetime of a single condensate [53] and could ultimately approach the standard quantum limit or beyond [53–55]. [...unchanged text...] [20–24,57,58]. Our work could inspire a new class of such quantum sensors.”***

The paper is well written and carefully edited and conveys the experimental techniques and analysis concisely. Important technical details are well presented in the supplemental information.

13) Thank you.

A few specific remarks:

a. In my observation of the field, the goal of CW atom lasers was prominently pursued over decades with collisional cooling of cold atoms beams. This work should be more prominently cited since it shares the vision with the current work. This work was carried out by three groups (Paris, Utrecht, Michigan), but only the Paris group is cited. I know that the Utrecht work has not been published in regular journals, but it would be fair to include it (via conference proceeding of the Dutch annual meeting, or the 2011 Ph.D. thesis).

14) We thank the referee for their suggestion and have included the noted references.

We changed old L74

“Great efforts were spent developing continuously cooled beams of atoms [31, 32] and...”

to the new L65

“Great efforts were spent developing continuously cooled beams of atoms [31, **Phys. Rev. A 73, 033622 (2006)**, Louise Kindt Ph.D. thesis Utrecht (2011), 32] and...”

b. The authors emphasize that the transparency beam increased the condensate lifetime from 40 ms to 1.5 s, and present this as important for their achievements. However, under CW operation, the lifetime of the condensate due to three-body collisions is shortened to 30 ms. This suggests that a much less stringent requirement for the transparency beam would have been sufficient.

15) The referee is correct, the condensate lifetime due to three-body collisions is on the order of 30ms (condensate size of 10^4 atoms with a flux of 2.4×10^5 atoms/s), which is much shorter than the 1.5s lifetime of an isolated condensate at the bottom of the dimple trap. It is also true that there may be some margin with respect to the degree of protection provided by the transparency beam, but probably much less than one might expect simply by comparing these lifetimes. The accumulation of atoms in the dimple is substantially reduced and no BEC is formed if only one transparency beam frequency component is present, or only 1/3 of the nominal transparency beam power is applied. We suspect several factors at play. Firstly, the protection must be provided throughout the dimple region including at the edges. Secondly, near the edges of the transparency beam the light shift acts to blue detune atoms, creating a heating zone. This zone can be minimized by using a large light shift and hence a large light shift gradient to minimize the width of this heating zone. Finally, the system is not linear, even small decreases in density or increases in temperature can have an outsized impact because we rely on collisions to load the dimple with atoms from the reservoir. Investigating these factors and finding simpler ways to obtain CW condensation is an interesting future research topic.

In the Methods section “Characterizing the transparency beam protection” we added, starting in L908 (former L819),

“No BEC is formed if only one transparency beam frequency component is present or only 1/3 of the nominal transparency beam power is applied.”

c. Figure caption 2: Figures a and b show two rectangles. Which one is the region of interest for figures c and d?

16) Thank you. We have modified the caption text of Fig. 2. as follows:

*“c, d, Optical density within the **rectangles** marked by corners in **a** and **b**, averaged along y. Fitted profiles using a thermal-only distribution (green dashed line) or a bi-modal distribution,...”*

To

*“c, d, Optical density within the **elongated rectangles** marked by corners in **a** and **b**, averaged along y. Fitted profiles using a thermal-only distribution (green dashed line) or a bi-modal distribution,...”*

Additional changes

While working on the resubmission we noticed a few small mistakes that we now corrected.

A1) The order of appearance of Extended Data Fig. 1 and 2 was swapped to align it with the order in which these figures are referred to in the text.

A2) We corrected a typo in the caption of Extended Data Fig. 7. In the last sentence of the caption we now reference Fig. 3 instead of Fig. 2. We also clarified this reference. Instead of
*“The phase-space density is estimated from measurements of the atom number and temperature in the dimple, **as detailed in Fig. 2**”*

we write

*“The phase-space density is estimated from measurements of the atom number and temperature in the dimple **in the same way as for Fig. 3.**”*

A3) We corrected a recurring typo. In the captions of Extended Data Fig. 2 (formerly 1), 4 and 7 we now write ρ_D instead of ρ .

A4) We added the meaning of the error bars in the caption of Extended Data Fig.3 and updated the Editorial Policy Checklist. The last sentence of the caption is now

“The error bars indicate estimates of the ranges in which the light shifted spectral lines lie.”

Reviewer Reports on the First Revision:

Referees' comments:

Referee #1 (Remarks to the Author):

The authors amended their manuscript following all the suggestions put forward in the previous review round. They considered the criticism raised especially concerning the relevance of their result, and now put their work in the right context, avoiding overstatements, citing previous pertinent results (e.g. continuous atom interferometry) and providing a more realistic conclusion section. As a consequence, I have no more reservations and fully recommend this manuscript for publication in Nature.

Referee #2 (Remarks to the Author):

The authors have carefully responded to the referees' comments and made changes to the manuscript which have addressed the main concern of both referees. The authors now clearly distinguish between short term possibilities of a CW BEC, conceptual studies and possible improvements of precision measurements. The claims are no longer overstated.

I want to commend the authors for the detailed response and the marked-up versions of the manuscript. This made it much easier for the referees to recognize the modifications to the manuscript.

I think the manuscript is still missing at least a short discussion of coherence times and limitations. After all, the most important qualitatively new feature of CW lasers was the huge improvement in coherence time. In line 226, the authors mention the improvement of matter wave coherence, and feature this also in the outlook (lines 265-270). The authors should make it clear that a CW BEC is removing one possible limitation of coherence times due to the pulsed operation. However, coherence times of 1 second or longer will require extreme stability of atom densities (maybe provide a number, 0.1% or so) and external fields, including laser intensities (due to AC Stark shifts, at the level of xyz).

With this addition, I recommend publication in Nature.

Author Rebuttals to First Revision:

Response to referees

We thank both referees for analyzing our modified manuscript and for their recommendation to publish in Nature. In the following we respond individually to the referees comments (blue, italic).

Referee #1 (Remarks to the Author):

The authors amended their manuscript following all the suggestions put forward in the previous review round. They considered the criticism raised especially concerning the relevance of their result, and now put their work in the right context, avoiding overstatements, citing previous pertinent results (e.g. continuous atom interferometry) and providing a more realistic conclusion section. As a consequence, I have no more reservations and fully recommend this manuscript for publication in Nature.

Thank you!

No changes to the manuscript were requested by Referee #1.

Referee #2 (Remarks to the Author):

The authors have carefully responded to the referees' comments and made changes to the manuscript which have addressed the main concern of both referees. The authors now clearly distinguish between short term possibilities of a CW BEC, conceptual studies and possible improvements of precision measurements. The claims are no longer overstated.

I want to commend the authors for the detailed response and the marked-up versions of the manuscript. This made is much easier for the referees to recognize the modifications to the manuscript.

*I think the manuscript is still missing at least a short discussion of coherence times and limitations. After all, the most important qualitatively new feature of CW lasers was the huge improvement in coherence time. In line 226, the authors mention the improvement of matter wave coherence, and feature this also in the outlook (lines 265-270). The authors should make it clear that a CW BEC is removing one possible limitation of coherence times due to the pulsed operation. However, coherence times of 1 second or longer will require extreme stability of atom densities (maybe provide a number, 0.1% or so) and external fields, including laser intensities (due to AC Stark shifts, at the level of xyz).
With this addition, I recommend publication in Nature.*

Thank you!

We included nearly all comments of referee #2 by changing the text starting line number 264
“Techniques like feedback could be used to overcome many sources of noise and stabilize the CW BEC atom number and phase [51, 52 (now 52,53)]. In the long term, matter-wave coherence will no longer be fundamentally limited by the atom number and lifetime of a single condensate [53 (now 51)] and could ultimately approach the standard quantum limit or beyond [53-55 (now 51,54,55)].”

into

“A CW BEC allows overcoming limits imposed on matter-wave coherence by the finite lifetime and atom number of a single condensate [51]. In practice, exceeding this limit will require extreme field stability, including external fields like dipole trap laser fields and the condensate mean field. For

example, a coherence time exceeding 1s requires an atom number stability on the order of 0.1%. Techniques like feedback could be used to overcome such sources of noise [52,53] and could ultimately allow coherence approaching the standard quantum limit or beyond [51,54,55].”

We considered the referee’s suggestion to include numbers for the required atom number and dipole trap depth stability. Both numbers depend on the exact conditions of BEC preparation and outcoupling. We now state the required atom number stability in our situation, but refrained from including an estimation of the dipole trap depth stability since it even more strongly depends on the exact method used to prepare the BEC and outcouple an atom laser beam.

Additional changes

- During the writing of this manuscript some team members were paid by grants that so far were not acknowledged. We added acknowledgments to those grants. We also added the reference number of C.-C. C’s scholarship.

Old text:

“We thank the NWO for funding through Vici grant No. 680-47-619 and the European Research Council (ERC) for funding under Project No. 615117 QuantStro. This project has received funding from the European Union’s Horizon 2020 research and innovation programme under grant agreement No 820404 (iqClock project). B.P. thanks the NWO for funding through Veni grant No. 680-47-438 and C.-C.C. thanks support from the MOE Technologies Incubation Scholarship from the Taiwan Ministry of Education.”

New text:

“We thank the NWO for funding through Vici grant No. 680-47-619 and NWA Startimpuls 2 grant NWA.QUANTUMNANO.2019.002. We thank the European Research Council (ERC) for funding under Project No. 615117 QuantStro. This project has received funding from the European Union’s Horizon 2020 research and innovation programme under grant agreement No 820404 (iqClock project). This work was financially supported by the Dutch Ministry of Economic Affairs and Climate Policy (EZK), as part of the Quantum Delta NL programme. B.P. thanks the NWO for funding through Veni grant No. 680-47-438 and C.-C.C. thanks Taiwan’s Ministry of Education for MOE Technologies Incubation Scholarship no. 60010200068.”

- Extended Data Table I: Changed “Under “Detuning” ...” to “In column “Detuning”, ...” in order to make the sentence more readable at its final place, under the table.
- Figure 1: label “Laser beams spatial extent” -> “Spatial extent of laser beams”
- Extended Data Figure 3: label “Operational intensity” -> “Operating intensity”; also changed in caption.
- Extended Data Fig. 6 was only cited in the Supplementary Information. As requested we now also cite it at an appropriate place in the main text (line 196):
“The pronounced anisotropic shape of the BEC in Fig. 2f is consistent with the expansion of a BEC from the anisotropic dimple, whose trap frequency along the y axis is approximately double that along z (see Supplementary Information and Extended Data Fig. 6).”

- *Added missing hyphen in line 650: "a right circularly polarized 1070 nm laser beam" -> "a right circularly polarized 1070-nm laser beam"*